# Favorable Biological Performance Regarding the Interaction between Gold Nanoparticles and Mesenchymal Stem Cells

**DOI:** 10.3390/ijms24010005

**Published:** 2022-12-20

**Authors:** Ruei-Hong Lin, Hsu-Tung Lee, Chun-An Yeh, Yi-Chin Yang, Chiung-Chyi Shen, Kai-Bo Chang, Bai-Shuan Liu, Hsien-Hsu Hsieh, Hui-Min David Wang, Huey-Shan Hung

**Affiliations:** 1Department of Neurosurgery, Neurological Institute, Taichung Veterans General Hospital, Taichung 40705, Taiwan; 2Program in Tissue Engineering and Regenerative Medicine, National Chung Hsing University, Taichung 40227, Taiwan; 3Graduate Institute of Biomedical Science, China Medical University, Taichung 40402, Taiwan; 4Department of Medical Imaging and Radiological Sciences, Central Taiwan University of Science and Technology, Taichung 40601, Taiwan; 5Blood Bank, Taichung Veterans General Hospital, Taichung 40705, Taiwan; 6Graduate Institute of Medicine, College of Medicine, Kaohsiung Medical University, Kaohsiung 80708, Taiwan; 7Graduate Institute of Biomedical Engineering, National Chung Hsing University, Taichung 40227, Taiwan; 8Translational Medicine Research, China Medical University Hospital, Taichung 40402, Taiwan

**Keywords:** gold nanoparticle, mesenchymal stem cell, biocompatibility, regenerative medicine

## Abstract

Gold nanoparticles (AuNPs) are well known to interact with cells, leading to different cell behaviors such as cell proliferation and differentiation capacity. Biocompatibility and biological functions enhanced by nanomedicine are the most concerning factors in clinical approaches. In the present research, AuNP solutions were prepared at concentrations of 1.25, 2.5, 5 and 10 ppm for biocompatibility investigations. Ultraviolet–visible spectroscopy was applied to identify the presence of AuNPs under the various concentrations. Dynamic Light Scattering assay was used for the characterization of the size of the AuNPs. The shape of the AuNPs was observed through a Scanning Electron Microscope. Afterward, the mesenchymal stem cells (MSCs) were treated with a differentiation concentration of AuNP solutions in order to measure the biocompatibility of the nanoparticles. Our results demonstrate that AuNPs at 1.25 and 2.5 ppm could significantly enhance MSC proliferation, decrease reactive oxygen species (ROS) generation and attenuate platelet/monocyte activation. Furthermore, the MSC morphology was observed in the presence of filopodia and lamellipodia while being incubated with 1.25 and 2.5 ppm AuNPs, indicating that the adhesion ability was enhanced by the nanoparticles. The expression of matrix metalloproteinase (MMP-2/9) in MSCs was found to be more highly expressed under 1.25 and 2.5 ppm AuNP treatment, relating to better cell migrating ability. Additionally, the cell apoptosis of MSCs investigated with Annexin-V/PI double staining assay and the Fluorescence Activated Cell Sorting (FACS) method demonstrated the lower population of apoptotic cells in 1.25 and 2.5 ppm AuNP treatments, as compared to high concentrations of AuNPs. Additionally, results from a Western blotting assay explored the possibility that the anti-apoptotic proteins Cyclin-D1 and Bcl-2 were remarkably expressed. Meanwhile, real-time PCR analysis demonstrated that the 1.25 and 2.5 ppm AuNP solutions induced a lower expression of inflammatory cytokines (TNF-α, IL-1β, IFN-γ, IL-6 and IL-8). According to the tests performed on an animal model, AuNP 1.25 and 2.5 ppm treatments exhibited the better biocompatibility performance, including anti-inflammation and endothelialization. In brief, 1.25 and 2.5 ppm of AuNP solution was verified to strengthen the biological functions of MSCs, and thus suggests that AuNPs become the biocompatibility nanomedicine for regeneration research.

## 1. Introduction

Regenerative medicine has been investigated for decades in regard to its clinical applications, e.g., tissue engineering and stem-cell therapies [1]. The strategies of regenerative medicine focus on the repair of tissue and organs which are damaged by trauma or diseases, thus leading to loss of functions [2]. To achieve the necessary therapeutic effect for patients, the combination of biocompatible scaffolds and cell biology is the main aspect surrounding tissue healing [3]. However, conventional regenerative medicine has encountered its share of bottlenecks, including insufficient cell proliferation, as well as differentiation and lower mechanical properties of scaffolds [4,5]. Therefore, the concept of nanomaterials was introduced into the design of tissue engineering materials, possibly improving the toughness of scaffolds and strengthening the biofunctions of stem cells through the nanomaterials’ biomimetic properties [6]. In vascular tissue engineering, many limitations and drawbacks exist in the development of artificial blood vessels, which may cause hyperplasia, inflammation and tissue destruction [7]. Thus, the characteristics of biomimetic properties, mechanical strength and long-term regeneration for developing vascular scaffolds have been concerning [8]. A previous study has shown that nanogold composite film substrates can induce the differentiation of mesenchymal stem cells (MSCs) into vascular endothelial cells with better mechanical properties for the development of vascular scaffolds [9]. In bone tissue engineering, the limitation of osteogenesis was the insufficient number of bioactive molecules [10]. The literature has verified that nanoparticles can combine with bioactive molecules or growth factors for delivery to bone injury sites [11]. Moreover, the modification of hydroxyapatite on the nanocomposite films can help achieve osteoconduction for bone repair [12]. In neural tissue regeneration, the nanomaterials with an aligned surface have been proven to culture with neural cells [13]. Accordingly, the combination of regenerative medicine and nanotechnology can enhance the efficiency of clinical treatments.

Bulk gold possesses the properties of anti-corrosion, non-degradability and good ductility. While bulk gold becomes gold nanoparticles (AuNPs), the electron cloud on the AuNP surface can experience surface plasmon resonance (SPR) with light at a wavelength near 520 nm [14]. Additionally, gold nanoparticles possess various unique physicochemical properties such as tunable size and structure, which allow AuNPs to become widely studied nanoscale metallic materials [15]. There are two manufacturing processes for AuNPs: chemical and physical. Chemically manufactured AuNPs can be obtained through the reduction of tetrachloroauric acid using sodium citrate [16]. Another related research has demonstrated using sodium borohydride as a reducing agent in order to collect AuNPs [17]. The physically manufactured AuNPs are acquired through physical vapor deposition (PVD), epitaxial stacking and physical crushing methods, as well as others [18]. The nanogold particles can have various properties due to different manufacturing processes. Chemically manufactured AuNPs will retain chemical substances after oxidative reduction, which can easily cause cytotoxicity and side effects [19]. However, there are no chemical residues left on physically manufactured AuNPs, which can exhibit better purity and biosafety characteristics [20]. Consequently, we applied physically manufactured AuNPs for the biocompatibility investigations that were performed in the present research.

Gold nanoparticles (AuNPs) have been applied in various fields due to their unique physicochemical properties, including materials science and nanotechnology [21]. Furthermore, the good biocompatibility and low cytotoxicity of AuNPs allow them to be widely used in the field of biomedicine, including being used as drug delivery carriers, in bioimaging and nanocomposite substrates [21]. Various physicochemical properties of AuNPs were introduced to manage cell behavior, including shape, diameter and the surface charge/modification [22]. The AuNPs could be absorbed by cells via clathrin-mediated endocytosis and phagocytosis mechanisms, followed by the activation of serial pathways to regulate cell behavior [23]. Previous research has elucidated that AuNPs combined with fibronectin and collagen extracellular matrix (ECM) substrates can facilitate the proliferation and migration of mesenchymal stem cells for vascular repair [24,25]. In addition, it has been demonstrated that AuNPs combined with nanofibers can induce stem cell differentiation and myocardial regeneration after electrification, owing to the unique surface electrochemical properties of AuNPs [26]. Moreover, the cooperation of AuNPs with polyethylene glycol is beneficial for the treatment of early-stage spinal cord injury by enhancing the survival of neurons and the formation of the myelin sheath, as seen in mice models [27]. When discussing cancer therapy, AuNPs were used as nanocarriers for improving transfection efficiency in lung cancer cells without influencing the proliferation of MSCs [28]. AuNPs possess great potential for improving the biocompatibility of nanomaterial substrates and strengthening the biofunctions of MSCs for biomedical approaches.

Mesenchymal stem cells (MSCs) are “multipotent” and can differentiate into most cell types, including adipose, bone, muscle, nerve and endothelial cells under an environment containing specific growth factors [29]. In previous studies, MSCs could express higher CD31 (platelet endothelial cell adhesion molecule-1) and vWF (Von Willebrand factor) on nanocomposite substrates modified via AuNPs, indicating that MSCs were able to potentially differentiate into endothelial cells for purposes of vascular regeneration [9]. MSCs are considered to be important in the development of regenerative medicine due to their capacity for multi-lineage differentiation and tissue repair [29]. MSCs have been verified as hypoimmune cells after transplantation, and are capable of treating neurodegenerative diseases as well [30]. Furthermore, MSCs can be easily isolated from most tissues, including bone marrow, the umbilical cord and fat and skin tissues [29], and have been applied for use in various clinical treatments such as inflammatory bowel diseases and diabetes [31,32].

As stated above, in view of the advantages of physically manufactured AuNPs, the biocompatible AuNPs have been used for modification of biomaterials to achieve better biological efficiency [9,12,25]. In the present study, we applied MSCs to use as treatment with various concentrations of AuNPs in order to evaluate their biological performance, with expectations of determining an optimal AuNP concentration for enhancing drug delivery and tissue engineering efficiency in clinical approaches.

## 2. Results

### 2.1. Characterization of AuNP

Figure 1A represents the brief procedure undertaken for preparation of the AuNP solutions. The concentrations of AuNPs were prepared as four groups: 1.25, 2.5, 5 and 10 ppm. The control group contained 0 ppm of AuNPs. Afterwards, each as-prepared AuNP solution was subjected to different characterizations. A scanning electron microscope was applied to observe the AuNPs used in the present research (Figure 1B), while the AuNP diameter quantified via Image J software was indicated as 39 ± 3.5 nm (Figure 1C). Figure 1D demonstrates the histogram of particle size distribution measured by DLS assay, and the accurate diameter of the AuNPs was measured as 45 ± 3.2 nm (Figure 1E). Furthermore, Figure 1F shows the UV-Vis absorbance spectra of the various AuNP solutions. A wavelength of 520 nm was verified to determine the presence of AuNPs in each group, and a higher AuNP concentration was explored to have a stronger absorbance peak as well.

### 2.2. Examination of Biocompatibility

The various biocompatibility assessments of the AuNPs were executed. The cell viability percentage of the MSCs influenced by the various AuNP treatments at 24, 48 and 72 h was investigated and is demonstrated in Figure 2A, with the tabulation organized as Appendix A. Afterwards, the MSCs were incubated with different amounts of AuNPs for 24 and 48 h and afterwards subjected to intracellular ROS detection. In Figure 2B, the semi-quantified data at 24 h indicated AuNPs at 1.25 and 2.5 ppm stimulated the lowest production of ROS, while the similar trend also shared at 48 h in Figure 2C. The above results demonstrate that while MSCs were incubated with AuNP 1.25 and 2.5 ppm treatments, the proliferation and anti-oxidant capacity of MSCs was significantly strengthened.

The activation of platelet cells and monocytes triggered through the various concentrations of AuNPs was further explored at 24 h. In line with the cell morphology of platelets from the SEM images (Figure 3A), the round shape platelets were observed in both the 1.25 and 2.5 ppm AuNPs, which indicated a non-activated form. The flattened morphology (active form) of the platelets was found in the AuNP 10 ppm group. Moreover, the adhered platelet cells and activation degrees were also analyzed, which are represented in Figure 3C,D. The number of adhered platelets was found to have decreased in each treatment. Next, the results for platelet activation degree explored that the 1.25 and 2.5 ppm AuNPs induced the lowest activation. In addition, it is considered that any immune response would be triggered while the monocytes (~5 μm) converted into macrophages (~40 μm). The expression of CD68 macrophage marker in the monocytes was measured at 96 h for various treatments. The fluorescence images are displayed in Figure 3B, with the semi-quantitation being analyzed based on CD68 fluorescence intensity shown in Figure 3E. The results indicated the 1.25 and 2.5 ppm AuNPs triggered the lowest expression of CD68 in macrophage. Furthermore, the number of monocytes/macrophages and the monocyte conversion yield (%) were counted and are shown in Appendix A. Based on the conversion percentage, the 1.25 and 2.5 ppm AuNPs stimulated the lowest monocyte conversion. According to the serial assessments, we verified that AuNPs at the concentration of 1.25 and 2.5 ppm were the biocompatibility nanomaterials.

The MSCs adhesion ability stimulated by the as-prepared AuNP solutions were investigated. The F-actin of the MSCs was stained by rhodamine phalloidin after 8 and 24 h of AuNP treatments, with the fluorescence photos displayed in Figure 4A. In accordance with the fluorescence images, the MSCs morphology was clearly observed to have filopodia and lamellipodia in both the 1.25 and 2.5 ppm AuNPs, while the SEM images (Appendix A) also demonstrate a similar cell morphology. Moreover, the cell aspect ratio of length-to-width within the MSCs was calculated, which the results indicated were significantly higher when treated with 1.25 and 2.5 ppm AuNPs (Figure 4B). The detailed data for the area, length and width of the MSCs is displayed in Appendix A. The MSCs’ morphology was enhanced to be more elongated while being treated with 1.25 and 2.5 ppm AuNPs, demonstrating that both cell adhesion and migration capacity were significantly strengthened. The above evidence also elucidates that AuNPs at the concentration of 1.25 and 2.5 ppm is the better biocompatibility nanomaterial.

### 2.3. Biological Functions Induced by AuNP

The expressed level of MMP-2/9 protein in MSCs treated with various concentrations of AuNPs was investigated, as it was related to migration ability. The zymograms at 24 and 48 h are displayed in Figure 5A. The semi-quantified results based on intensity are demonstrated in Figure 5B,C, which accounted for the greater expression induced by 1.25 and 2.5 ppm AuNPs.

Additionally, the MSCs’ migration ability was enhanced by various concentrations of AuNPs as demonstrated in Figure 5D, with the moving distance being measured in Figure 5E. In accordance with the fluorescence images at 24 h, the MSCs migrated into the boundary were observed to be greater with 1.25 and 2.5 ppm AuNP induction. Meanwhile, it could be clearly observed in the images at the time point of 24 h that the MSCs migrated to the center with both treatments mentioned above. Afterward, we quantified the boundary moving distance influenced by each treatment with Image J 5.0 software (National Institutes of Health, Bethesda, MD, USA). The results verified that AuNPs at the 1.25 and 2.5 ppm concentration greatly induced the MMPs’ expression, which significantly strengthened the migratory ability of the MSCs.

### 2.4. Investigation of Cell Progress and Apoptosis

The biocompatible nanomaterial prevents the cells from apoptosis, which is concerning. Therefore, the cell cycle progress of MSCs influenced by AuNPs after 48 h was analyzed using the FACS method. The histograms of the cell cycle are displayed in Figure 6A. The cell population in different phases is quantified in Figure 6B. In line with the semi-quantifications, the cells at the Sub-G1 phase were slightly decreased in both 1.25 and 2.5 ppm AuNP treatments, while the population in the 10 ppm AuNPs was more than others. The cell progress at G0G1 quantification showed a slight decrease with the addition of 1.25 and 2.5 ppm AuNPs. Interestingly, we discovered that the cell population at the S phase was greater in 1.25 and 2.5 ppm AuNP addition with a significant difference, indicating the MSCs passed the G1 phase and entered the S phase associated with mitosis and proliferation efficiency. However, the AuNP 5 treatment did not facilitate more MSCs to enter the S phase, moreover, the decreased cell population was explored in the 10 ppm AuNP. For the G2M phase, the quantified data demonstrated a higher level in the 1.25 and 2.5 ppm AuNP groups. On the contrary, the lower MSCs population was found in 5 and 10 ppm treatments. The evidence elucidated that at the concentration of 1.25 and 2.5 ppm, AuNP could facilitate MSCs to undergo the S phase and enter the G2M cell cycle progress.

Next, the apoptosis of the MSCs after various AuNP treatments was also investigated via the FACS method. Figure 6C,D demonstrate the histograms and quantitative data after 48 h of treatments, respectively. We analyzed the quantitative results with the exploration of a higher Annexin-V positive cells population (presented as apoptotic cell) induced by AuNPs at the concentration of 5 and 10 ppm, while the similar trend also shared in the detection of PI positive cells (presented as necrotic cells). The results of viable MSCs demonstrated the decreased level with significant difference in AuNP 5 and 10 ppm. On the contrary, the treatment of AuNP 1.25 and 2.5 ppm did not induce higher Annexin-V positive and PI positive MSCs (Figure 6D). Discussed with the viable and dead MSCs stimulated by each treatment, the results elucidated that the 2.5 and 5 ppm AuNP treatments lead to less viable cells and more dead cells, while the 1.25 and 2.5 ppm AuNP induced the opposite conditions (Figure 6D). Furthermore, Figure 7 exhibits the images of apoptotic MSCs and quantitative ratio through Annexin-V FITC/PI double staining assay. Figure 7A shows that the green fluorescence of the Annexin-V probe could be clearly observed in the 5 ppm and 10 ppm AuNP treatments, which indicates there were more apoptotic cells. We further semi-quantified the green fluorescence intensity of the Annexin-V probe in each AuNP addition, which demonstrates the higher ratio seen in both the 5 ppm AuNPs and 10 ppm AuNPs, particularly for 10 ppm AuNPs with significant difference (Figure 7B).

We subjected the MSCs to the investigation of apoptotic related protein expression after 48 h of incubation through Western blotting assay. The protein zymography of various proteins is shown in Figure 8A, with the quantitative results of each protein analyzed in Figure 8B–F. In accordance with the data, the apoptotic relative proteins, including Active-caspase-3, p21and Bax, were determined to have significantly lower expression in both the 1.25 and 2.5 ppm AuNP treatments. We explored that the treatment of 5 and 10 ppm AuNPs induced a greater apoptotic proteins expression, particularly for 10 ppm AuNP s with significant difference. On the contrary, the apoptosis inhibited protein Bcl-2 and cell cycle progress regulator protein Cyclin D1 were investigated to be remarkably expressed in MSCs in both the 1.25 and 2.5 ppm AuNP treatments. Additionally, in the treatment of 5 ppm AuNPs, both the Bcl-2 and Cyclin D1 expression were slightly decreased, while the 10 ppm AuNP group showed even lower expression with significant difference. Thus, we considered that AuNPs at both 1.25 and 2.5 ppm could enhance cell proliferation and induce lower apoptotic MSCs.

### 2.5. Expression of Inflammatory Cytokine Induced by AuNP

To investigate the inflammatory response stimulated by various concentrations of AuNPs, we processed real-time PCR analysis to realize the mRNA expression level in the MSCs. The cytokines including Tumor Necrosis Factor-α (TNF-α), Interferon gamma (IFN-γ), Interleukin-1β (IL-1β), Interleukin-6 (IL-6), Interleukin-8 (IL-8) and Interleukin-10 (IL-10) were chosen. In line with each quantification in Figure 9, the inflammatory cytokines, such as TNF-α and IFN-γ, were found to be lower expressed with significant difference in the 1.25 and 2.5 ppm AuNP treatments at each time point, while the mRNA expression of IL-1β, IL-6 and IL-8 also shared similar conditions. On the contrary, the expression level of anti-inflammatory cytokine IL-10 was explored to be significantly up-regulated in MSCs with treatments of 1.25 and 2.5 ppm AuNPs. The above evidence elucidates that AuNPs at the concentrations of 1.25 and 2.5 ppm have the ability to bring about anti-inflammation.

### 2.6. Biocompatibility in Animal Models

Biocompatibility assessments of nanomaterials including chronic inflammation have been a concern for tissue regeneration efficacy. Figure 10 and Figure 11 demonstrate the serial histology examinations and IHC staining assays via a rat model. Figure 10A shows the tissue images explored through Masson’s trichrome staining assay, with the blue color presented as collagen deposition. Figure 10D describes the quantitative results of collagen deposition, which indicated the lower deposition with significant difference after 1.25 and 2.5 ppm AuNP implantation. The H&E staining histological images of capsule formation were demonstrated in Figure 10B. The white arrows locate the capsule thickness in the tissue of each treatment. The thickness was semi-quantified and was shown in Figure 10E, which indicated the trend similar to collagen deposition. Next, the expression of a CD31 endothelialization marker was also evaluated, with the fluorescence images and semi-quantified data depicted in Figure 10C,F. The results demonstrate that 1.25 and 2.5 ppm AuNPs significantly induced endothelialization in tissue.

Moreover, the macrophage polarization influenced by AuNP treatments was investigated, where the CD86 marker was the M1 polarization and the CD163 marker was selected for M2 polarization. The fluorescence images for CD86 observation are displayed in Figure 11A, while the detection of CD163 is exhibited in Figure 11B. Based on the fluorescence images, we explored that the red color of CD86 was faded in both 1.25 and 2.5 ppm AuNP implantation. Otherwise, the green fluorescence of CD163 was stronger in AuNP 1.25 and 2.5 ppm compared to the others. To quantify the results, the fluorescence intensity of CD86/163 was measured and are shown in Figure 11D,E. The result of the CD86 intensity indicates the decreased expression with significant difference influenced by 1.25 and 2.5 ppm AuNPs. However, the reverse tendency was discovered in the semi-quantification of the CD163 marker intensity, which was determined to be significantly greater treated with 1.25 and 2.5 ppm AuNPs. Furthermore, the expression of the CD45 marker has also been a concern during the research when determining the leukocyte filtration (Figure 11C). The quantitative data was analyzed in Figure 11F. In line with the images and quantitative result of the CD45 marker expression, the lower fluorescence intensity was explored in each treatment, particularly for the 1.25 and 2.5 ppm AuNP implantations. Thus, with the evidence seen in the histology and IHC results, we determine that the 1.25 and 2.5 ppm AuNP treatments could significantly inhibit capsule formation, collagen deposition and inflammation response. The enhancement of endothelialization and M2 macrophage polarization suggests that 1.25 and 2.5 ppm AuNPs could facilitate tissue regeneration.

Figure 12 provides the illustration which concludes the biological performance of AuNPs. AuNPs at the concentrations of 1.25 and 2.5 ppm exhibited better biocompatibility and better induction of biological functions. Furthermore, 1.25 and 2.5 ppm AuNPs can prevent MSCs from apoptosis. The above evidence suggests 1.25 and 2.5 ppm AuNPs are fascinating nanomaterials for biomedical approaches.

## 3. Discussion

Conventional tissue engineering has its drawbacks, such as insufficient availability of appropriate organs/tissue, which, therefore, may require longer times for treatment or possibly cause severe disease in patients [33]. For instance, skin grafting approaches such as autografting and allografting will encounter the problems mentioned above. Additionally, a lack of donor skin tissue and immunological rejection can decrease the survival rate of patients [34]. Therefore, extracellular matrices (ECMs), e.g., collagen, fibronectin and chitosan, modified by metallic nanoparticles become attractive wound dressing materials, and the substitutes comprising MSCs and nanoparticle-modified composites are considered to be effective for tissue repair [35]. According to the previous literature, collagen modified with silver nanoparticles demonstrated the capacity for MSC endothelial differentiation in vascular regeneration [36], while the combination of fibronectin with silver nanoparticles also indicated good biological performance as well [37]. Furthermore, the pullulan-collagen-AuNPs [38] and chitosan-AuNP nanocomposite films [39] also demonstrated a neural differentiation capacity in MSCs with a better expression of neural-related markers (GFAP, β-Tubulin, and nestin), while also exhibiting good biocompatibility through both in vitro and in vivo assessments.

In regenerative medicine and tissue engineering, the developed biomaterials must be strictly examined for further clinical approaches, as biocompatibility is the most important indicator for clinical applications. The ultimate goal is to develop biomedical nanomaterials with low immunogenicity and better biofunction performance. Various approaches for synthesis of AuNPs including chemical and physical techniques have been investigated for decades [40]. The chemical synthesis of AuNPs can be easily functionalized for specific applications [41]. However, the usage of a reducing agent to acquire AuNPs and the functionalization via organic solvents and reagents are high-priced and toxic for biomedical applications and environments [42]. Moreover, the production of AuNPs via chemical process is low, which causes limitation for investigations [43]. In comparison with chemical manufacturing methods, the physical synthesis of AuNPs is acquired from the physical vapor deposition technique without addition of surfactant, stabilizer and reducing agents, which are characterized to be a controlled size at nano-scale, high surface area to volume ratio and superior biocompatibility with low toxicity for targeted drug delivery applications [40,44]. Thus, the physical synthesis of spherical AuNPs was applied for investigations of biological performances within MSCs.

In the present research, platelet activation and monocyte conversion ratios were investigated, which were demonstrated to be the lowest in the 1.25 and 2.5 ppm AuNP treatments. Based on the previous study, the activated platelet cells would adhere and become a dendritic form, while the aggregated platelet cells may lead to thrombosis [45]. Additionally, the inflammatory response would be initiated by the foreign body. When the immune response occurred, the monocytes differentiated into macrophages to attack the foreign particle [46]. Our results demonstrated that 1.25 and 2.5 ppm of AuNPs induced the lowest activation of platelet cells and monocytes, indicating that the concentration could be accepted by the immune system. Additionally, the immune response could also be regulated by the inflammatory cytokines, including TNF-α, IFN-γ, IL-1β, IL-6 and IL-8 [47]. We also explored the IL-10 anti-inflammatory cytokines [48] after AuNP treatments, where the expression of inflammatory cytokines was found to be lower in the MSCs in 1.25 and 2.5 ppm AuNPs, while the IL-10 had been upregulated. Furthermore, our previous studies investigated the apoptotic ratio with AuNP treatments. The AuNP-based nano-drug induced cytotoxicity in the A549 lung cancer cell line, but there was no significant inhibition on normal BEAS-2B cells [20]. According to results from the present study, we have determined that at the concentrations of 1.25 and 2.5 ppm of AuNPs, apoptotic MSCs were remarkably lower than 5 ppm and 10 ppm, respectively. Furthermore, the cell cycle histograms also indicated a higher percentage in the G0G1 and S phase [49], which demonstrated that MSC proliferation was induced by an appropriate concentration of AuNPs. The apoptotic-related proteins being regulated by gold nanoparticles was also a concern. The biocompatible nanomaterials would induce the expression of anti-apoptotic proteins [50]. Our evidence demonstrated the higher expression of Bcl-2 and Cyclin-D1, which has been indicated as the anti-apoptosis and cell cycle regulator. The evidence verified that AuNPs at the concentrations of 1.25 and 2.5 ppm displayed better biocompatibility for further applications.

Biofunctions of mesenchymal stem cells enhanced by appropriate nanomaterials play a critical role in tissue regeneration, with the regeneration rate being associated with the MSCs’ proliferation and migration ability [51]. As the literature reported, the SDF-1/CXCR4 axis initiates cell signaling pathways in cells associated with secreting matrix metalloproteinases (MMPs) and the vascular endothelial growth factor (VEGF) [52,53]. Afterward, the MMPs would be expressed in cells to degrade ECM that is associated with cell migration and vascular remodeling [54], while the MMP-2 and MMP-9 proteins could activate α5β3 integrin to enhance migratory ability and angiogenesis for tissue repair [55]. Previous research has indicated that AuNP-modified nanocomposites could facilitate the expression of MMPs and migratory distance as well [12]. Our results demonstrated that 1.25 and 2.5 ppm AuNP treatments had better MMP-2/9 expression and that the morphology of MSCs (filopodia and lamellipodia) was also induced, indicating that the adhesion and migratory ability was being enhanced. However, in the subsequent biological function tests, 5 to 10 ppm of nanogold particles was found to cause a downregulation in the biological function of mesenchymal stem cells. The literature also indicates that an excessive intake of nanogold particles leads to apoptosis [56]. Thus, 1.25 to 2.5 ppm of AuNPs may be the appropriate concentration for future modified nanomaterials to have homogenous morphology for inducing better adhesion and MMPs’ secretion of mesenchymal stem cells.

Regarding the results of in vivo biocompatibility measurements, the foreign body responses, including capsule formation and collagen deposition, were explored to be lower with 1.25 and 2.5 ppm AuNP treatments. Previous research also discussed the natural/synthetic polymers modified with an optimal amount of AuNPs that remarkably decreased foreign body responses [57]. Additionally, the stimulation of M1/M2 macrophage polarization was performed to be the significant induction of M2 macrophage for tissue pair by AuNPs [58]. The high intensity of M2 (CD163) macrophage and lower leukocyte (CD45) filtration in the present study elucidated 1.25 and 2.5 ppm AuNPs triggered lower inflammations, indicating the AuNPs can be a potential nanomedicine. However, the literature reviewed that the surface modification techniques could eliminate the limitations of functionalities among AuNPs to induce better biological responses [59]. As mentioned in a review report, the conventional approaches to the synthesis of AuNPs combined with biomaterials may decrease durability and cause toxicity or side effects that reduce therapeutic efficiency [60]. Thus, the physically synthesized AuNPs at an appropriate concentration in this study were a promising candidate for biomedical applications. While, the particle surface could be functionalized by biopolymers such as collagen or polyethylene glycol, which were applied as the biocompatible nano-drug delivery system with superior stability [20,61]. In accordance with this study, the physically synthesized AuNPs fabricated with polymeric biomaterials have more potential for clinical applications with good biosafety owing to the minimal toxicity effects without using reducing agents. Moreover, the AuNPs functionalized to carry various biomolecules can be manufactured for investigations of facilitating specific differentiation pathways in MSCs with better tissue repair.

## 4. Materials and Methods

### 4.1. Preparation of Gold Nanoparticle (AuNP) Solution

The bulk gold was vaporized to atomic level via an electrically gasified method under a vacuum. Next, the AuNPs were collected through a cold trap followed by centrifugation. The AuNPs resuspended in distilled water contained no surfactant. The size of the AuNPs could be managed by evaporation time and electric current settings [62]. The concentration of the AuNP stock solution was 50 ppm (GOLD NANOTECH, INC., Taipei, Taiwan), which was filtered using a 0.22 μm filter. Therefore, the solution containing AuNPs was considered to be sterile. Next, the AuNP solution was diluted with a culture medium calculated through the mass conservation equation: M1V1 = M2V2 (“M” being the concentration of solution, “V” being the volume of solution). Afterward, the various concentrations of the AuNP solution at 1.25, 2.5, 5 and 10 ppm were prepared. In the present research, the cells were seeded on cover-slips and 6-well, 24-well and 96-well 10 cm culture dishes prior to the addition of the AuNP solutions. The Control groups (0 ppm of AuNPs) were treated only with the culture medium. The above as-prepared materials were applied for further in vitro experiments.

### 4.2. Material Characterization

Ultraviolet–visible (UV-Vis) spectra of the AuNPs were obtained from a Helios Zeta spectrophotometer (Thermo Fisher, Pittsfield, MA, USA), with the wave range being from 400 to 650 nm. The quartz cuvette was washed with deionized water (ddH_2_O) and cautiously wiped with mirror paper prior to the addition of ddH_2_O for background absorbance. Next, the as-prepared sample was characterized sequentially, while the cuvette was cleaned with ddH_2_O between each measurement in order to remove any residual solution. Origin Pro 8 software (Originlab Corporation, Northampton, MA, USA) was used to analyze the obtained data.

Dynamic light scattering (DLS) assay was processed through a Zetasizer Nano ZS instrument (Malvern Panalytical Ltd., Malvern, Worcestershire, UK) for examination of the AuNP hydrodynamic diameters. One mL of each as-prepared material was added into a quartz cuvette for measurement which was executed under a 532 nm light source with a 90-degree scattering angle.

A Scanning Electron Microscope (SEM, JEOL JEM-5200, JEOL Ltd., Tokyo, Japan) was used to observe the shape of the AuNPs. The AuNP solution was loaded on a silicon wafer and dried out at 80 °C. Next, the wafer containing the AuNPs was sputter coated with silver in order to explore the structure. The voltage was 5.0 kV, while the scale bar was 1 μm. The data obtained from the experiments mentioned above were measured in triplicate.

### 4.3. Cell Culture

The mesenchymal stem cells (MSCs) were collected from human umbilical cord Wharton’s jelly tissue, which was described in our previous research [36]. An H-DMEM culture medium containing 10% fetal bovine serum (FBS) and 1% (*v/v*) penicillin/streptomycin antibiotics (100 U/mL) was applied for MSC incubation at 37 °C with a 5% CO_2_ atmosphere. The MSCs for various assessments in this research were at the 8th passage.

### 4.4. Biocompatibility Assessments

#### 4.4.1. Cell Proliferation and Reactive Oxygen Species (ROS) Measurements

We used 3-(4,5-dimethylthiazol-2-yl)-2,5-diphenyl tetrazolium bromide (MTT, Sigma-Aldrich, St. Louis, MO, USA) to detect the MSC proliferation. MSCs (1 × 10^4^ cells/well) were first cultured in a 96-well culture plate at 37 °C with a 5% CO_2_ atmosphere. After 24 h for cell attachment, different concentrations of AuNP solution (1.25, 2.5, 5 and 10 ppm) were loaded into the culture plate. After being incubated with various materials for 24, 48 and 72 h, 100 μL of MTT solution (0.5 mg/mL) was added for 2 h of incubation (37 °C, 5% CO_2_). Next, 100 μL of Dimethyl sulfoxide (DMSO) solution was added for an extra 10 min of incubation. A SpectraMax M2 ELISA reader (Molecular Devices, San Jose, CA, USA) was applied to detect the absorbance at 570 nm.

Intercellular ROS production in MSCs with different treatments was detected by a DCFH-DA (2′,7′-dichlorofluorescin diacetate) fluorescent probe (Sigma-Aldrich, USA) [24]. The MSCs (2 × 10^5^ cells/well) were seeded into a 6-well culture plate (37 °C, 5% CO_2_) to stand for cell attachment after 24 h. Next, the as-prepared AuNP solutions were loaded into the culture plate for 24 and 48 h of incubation. After the incubation, a 0.05% Trypsin-EDTA solution was added to harvest the MSCs which were then washed with PBS. A DCFH-DA (10 nM) solution was used to detect ROS in a dark room for a period of 30 min. The fluorescein-positive cells were then examined by a flow cytometer and quantified using Flow Jo Version 7.6 software (Becton Dickinson, Canton, MA, USA).

#### 4.4.2. Platelet and Monocyte Activation Test

Both human platelet cells and monocytes were collected by following Percoll protocol, with IRB approval being given from Taichung Veterans Hospital (CE12164). The monocytes were stored in an RPMI culture medium with 10% FBS and 1% (*v/v*) antibiotics (10,000 U/mL penicillin G and 10 mg/mL streptomycin). Next, the monocytes (1 × 10^5^ cells/well) were added into a 24-well culture plate along with the addition of various as-prepared AuNP solutions for 96 h of incubation (37 °C, 5% CO_2_). Afterward, the cells were collected by trypsin. The convert ratio was observed and calculated through a microscope. Additionally, to further investigate the inflammatory response induced by AuNP solutions, the anti-CD68 (macrophage marker) primary antibody (GeneTex Incorporation, Irvine, CA, USA) and FITC secondary antibodies were applied by using the immunofluorescence staining (IF) method.

The platelet cells (2 × 10^6^ cells/well) were cultured with an as-prepared AuNP solution for 24 h at 37 °C within a 5% CO_2_ humified atmosphere. Next, the cells were fixed by a 2.5% glutaraldehyde solution for eight hours, washed twice with PBS, and dehydrated through the adding of 30% to 100% alcohol. The platelet cells were dried out using a Critical Point Dryer (CPD), and the morphology was scanned by SEM (JEOL Ltd., Japan). To identify the platelet activation degree, the standard equation was used, described as followed: 0 = round form (non-activated), 1/4 = dendritic form (pseudopodial with no flattened), 1/2 = spread-dendritic form (flattened and pseudopodial type), 3/4 = spreading form (pseudopodial with hyalopasm spreading), and 1 = completely spread form (activated). The average activation was quantified according to 50 adhered platelets [25].

#### 4.4.3. Observation of Cell Adhesion

The MSCs (1 × 10^4^ cells/well) were seeded into 24-well culture plates with 15 mm glass coverslips and treated with AuNP solutions to observe the cytoskeleton. The culture plates containing cells were incubated for 8 and 24 h at 37 °C within a 5% CO_2_ atmosphere. The cells were fixed with 4% paraformaldehyde (PFA, Sigma-Aldrich, USA) at 4 °C for 30 min. After being washed thrice with PBS, a 0.5% Triton X-100 (Sigma-Aldrich, USA) solution was added for 10 min to bring about a reaction. Next, a 5% bovine serum albumin (BSA, Sigma-Aldrich) solution was added for blocking. Ultimately, the MSCs were stained by 6 μM rhodamine phalloidin (Sigma-Aldrich) in a dark room for 30 min at room temperature. The cell nuclei were located through the use of a 1 μg/mL 4,6-diamidion-2-phenylindole (DAPI, Invitrogen, Waltham, MA, USA) solution for a period of 10 min. The cytoskeleton was cautiously observed using a Zeiss Axio Imager A1 fluorescence microscope (Carl Zeiss AG, Jena, Germany). The size, length and width of the MSCs were measured using Image Pro Plus 5.0 software (Media Cybernetics, Rockville, MA, USA). The cell aspect ratio was measured by a length-to-width equation, which was described in a previous study [57]. Furthermore, the morphology of the MSCs adhesion treated with various as-prepared materials was also scanned with SEM. The experiments were processed in triplicate.

### 4.5. Investigation of Biological Functions

#### 4.5.1. Metalloproteinase Zymography Assay (MMP)

MSCs (2 × 10^5^ cells/well) were cultured in a 6-well culture plate and treated with various concentrations of AuNP solutions. After undergoing incubation for 24 and 48 h at 37 °C within a 5% CO_2_ atmosphere, the medium containing the MMP protein was collected for gelatin zymography assay [63]. In brief, the gel with protease digest bands could be clearly observed within the dark blue background. The gel was scanned and the expression of MMPs was semi-quantified using Gel-Pro Analyzer 4.0 software (Media Cybernetics, Burlington, MA, USA).

#### 4.5.2. Cell Migration Assay

The MSCs (1 × 10^4^ cells/well) were cultured with Oris^TM^ seeding stoppers (Platypus Technologies, Madison, WI, USA) in an incubator (37 °C within a 5% CO_2_ atmosphere) to reach confluency. Next, the seeding stoppers were removed to add the as-prepared AuNP solutions for 24 and 48 h of incubation. After the incubation periods, 2 μM of calcein-AM solution was added for 30 min of additional incubation (37 °C, 5% CO_2_). The cell migration ability of each treatment was displayed through a fluorescence microscope (Zeiss Axio Imager A1, Carl Zeiss AG, Jena, Germany). The boundary migration distance of the cells was analyzed with Image J 5.0 software (National Institutes of Health, Bethesda, MD, USA).

### 4.6. Investigation of Cell Cycle and Apoptosis

#### 4.6.1. Cell Cycle Analysis

MSCs (2 × 10^5^ cells/well) were cultured in a 6-well culture plate for 24 h under 37 °C with 5% CO_2_ incubation for cell attachment. Next, the MSCs were treated with different concentrations of AuNP solutions for 48 h of incubation, then collected through 0.05% trypsin-EDTA and washed twice with PBS solution. The cells were centrifuged at 4 °C, 1200–1500 rpm for 10 min. The supernatant was removed, and the cells then fixed with 75% alcohol at −20 °C for at least 8 h. After the cells were collected and washed with PBS, a propidium iodide (PI, Sigma-Aldrich, USA) solution was added to stain the cell nuclei for 30 min. The cell progress was detected and analyzed through a fluorescence-activated cell sorting (FACS) Calibur flow cytometer (BD Biosciences, Franklin Lakes, NJ, USA). The experiments were processed in triplicate.

#### 4.6.2. Cell Apoptosis

MSCs at a cell density of 2 × 10^5^ cells per well were cultured in a 6-well culture plate for 48 h of incubation (37 °C, 5% CO_2_). The apoptotic cells induced by the various concentrations of AuNP treatments were detected through Annexin-V/PI double staining assay (Sigma-Aldrich, Burlington, MA, USA). In brief, the Annexin-V solution was able to target the phosphatidylserine (PS) that was exposed on the extracellular face of the plasma membrane during the early stage of cell apoptosis. A PI solution was applied to detect the cell nuclei. The cells stained through Annexin-V/PI double staining were observed through a fluorescence microscope, with the fluorescence density semi-quantified via Image J 5.0 software (Media Cybernetics, Burlington, MA, USA). Furthermore, the apoptotic cells (Annexin V^+^/PI^−^), necrotic cells (Annexin V^−^/PI^+^) and dead cells (Annexin V^+^/PI^+^) were all investigated through a FACS Calibur flow cytometer (BD Biosciences, USA), and the data then analyzed using Flow Jo Version 7.6 software (Becton Dickinson). The experiments were processed in triplicate.

### 4.7. Western Blotting Assay

The MSCs at 2 × 10^5^ cells per well were cultured with various AuNP treatments in a 10 cm^2^ culture plate for 48 h of incubation (37 °C within a 5% CO_2_ atmosphere). The cells were harvested via a 0.05% trypsin-EDTA solution and washed twice with PBS. The RIPA buffer (pH value = 7.5, 5 mL of Tris, 3 mL of NaCl, 1 mL of NP-40, 1 mL of 10% SDS, 0.5 g of sodium deoxycholate) was added for 1 h in ice to lysis the cells. After the process, the lysis cells were centrifuged at 4 °C, 13,000 rpm for 20 min to obtain the supernatant containing proteins. Next, the concentration of the proteins was measured through the use of a BCA Protein Assay Reagent Kit by following the manufacturer’s instructions (Bio-Rad Laboratories Inc, Hercules, CA, USA). Afterward, each protein sample was electrophoresed in an SDS-PAGE gel at 120 Voltage for 90–120 min to separate the proteins. The transfer buffer (3 g of Tris base, 14.4 g of glycine, 150 mL of 100% methanol, 1000 mL of ddH_2_O) was added into the device (Bio-Rad Mini Protean System) to transfer the protein onto the PVDF membrane (Immobilon P; EMD Millipore) in ice for 1 h at 400 mA. The membrane containing proteins was blocked in TBST (60.57 g of Tris-HCl, 85 g of NaCl, 5 mL of Tween-20, 1000 mL of ddH_2_O) with 5% milk powder for 1 h after shaking, then probed with various primary antibodies (Santa Cruz, CA, USA) at 4 °C overnight [Caspase-3, p21, Bax, Bcl-2, Cyclin-D1 at a 1:1000 dilution, and β-actin at a 1:2000 dilution]. Next, the membrane was washed thrice with TBST solution and incubated with a 1:2000 dilution of HRP-conjugated goat anti-rabbit or anti-mice IgG (Zhongshan Goldenbridge Biotechnology, China) for 1 h at room temperature. The immunoblots were detected via an ECL kit (PerkinElmer, Waltham, MA, USA) and the protein bands were observed under X-ray. Each protein expression was measured by a Gel-Pro Analyzer 4.0 (Media Cybernetics, USA), with the band density normalized to β-actin. The experiments were processed in triplicate.

### 4.8. Real-Time Polymerase Chain Reaction (PCR) Analysis

The MSCs at the density of 2 × 10^5^ cells per well were cultured in a 6-well culture plate for 24 h to stand for cell attachment. Afterward, the cells were treated with various concentrations of AuNP solutions for 4, 8, 12 and 24 h. The total amount of mRNA was extracted by following the manufacturer’s instructions. The collected cells were treated with 1 mL of TRIzol (Invitrogen, USA) for 5 min, then added to 200 μL of chloroform (Sigma, USA) for 15 s of shaking, before standing for 3 min at room temperature to extract the RNA. The samples were centrifuged for 15 min at 4 °C, 12,000 rpm. Next, we removed the supernatant, added 0.5 mL of isopropanol for 10 min of incubation at 4 °C, and then centrifuged for 15 min at 4 °C. The supernatant was removed and washed twice with 1 mL of 75% alcohol. The samples with RNA were dried out and restored by 20 μL DEPC-treated H_2_O. The concentration of each RNA sample was measured at 260 nm with a SpectraMax M2 ELISA reader (Molecular Devices, San Jose, CA, USA).

cDNA synthesis was processed with a RevertAid™ First Strand cDNA Synthesis Kit (Thermo Fisher, USA) based on the manufacturer’s instructions. The 2 μL of oligo (dT)_18_ primer and random hexamers (1:1) were loaded into RNA samples, then placed into a gradient polymerase reaction temperature controller (Life Technologies, Carlsbad, CA, USA) at a setting of 65 °C for a period of 5 min. Next, the mixture containing 1 μL of Lock^TM^ RNase inhibitor (20 U/mL), 2 μL of dNTP Mix (10 mM), 4 μL of 5× reaction buffer and 1 μL of RevertAid™ M-MuLV Reverse Transcriptase (200 U/μL) was loaded into the samples for a 1-h reaction period at a temperature of 42 °C. Ultimately, the samples were reacted at 70 °C for 5 min, and the cDNA samples stored at −80 °C.

The IQ^2^ Fast qPCR System was applied to execute polymerase chain reaction (PCR) by using the cDNA as a template according to the manufacturer’s instructions. The 10 μL of total reaction volume contained 0.5 μL of primer (0.3 μM) and 5 μL of enzyme. Each mRNA expression was measured by the StepOnePlus™ Real-Time PCR System. The experiments were processed in triplicate.

### 4.9. Animal Model

In the present research, we prepared 2–3-month-old, 300–350 g Female Sprague Dawley (SD) rats based upon the China Medical University Animal Care and Use Committee’s approval (CMUIACUC-2021-074). The various concentrations of AuNP were firstly coated on both sides of 15 mm glass coverslips for 30 min in a 6-well culture plate. Next, the dorsal skin of the rats was incised at 10 mm × 10 mm to implant the various AuNP treatments after local anesthesia was given. After 4 weeks of subcutaneous implantation, the wound tissues surrounding with implanted AuNP were resected for histological and immunohistochemistry measurements. The specimens for various assessments were fixed with 10% formaldehyde, dehydrated with serial concentrations of ethanol, and embedded with paraffin. The specimens were longitudinally sectioned at 4 μm thickness [64]. The capsule formation was evaluated over 6 sites through hematoxylin and eosin (H&E) staining assay. Masson’s trichrome staining (Sigma-Aldrich, USA) assay was applied for the measurement of collagen deposition. The immunohistochemical staining (IHC) of the CD31 endothelial marker was detected by anti-rabbit primary antibodies (1:200 dilution, Sigma-Aldrich, USA). The M1/M2 macrophage polarization was examined by CD86/CD163 anti-mouse monoclonal antibodies (1:200 dilution, Santa Cruz, USA). Furthermore, the leukocyte filtration was measured by CD45 anti-mouse monoclonal antibodies (1:200 dilution, Santa Cruz, USA). The signal amplification was executed via the use of AF488 Donkey anti-mouse IgG secondary antibodies (1:500 dilution, Invitrogen, USA). A DAPI solution was used to target cell nuclei. The tissue was used for IHC staining. The fluorescence was observed through an Olympus ix71 fluorescence microscope (Tokyo, Japan), with intensity quantified by Image J 5.0 software. The results are presented as mean ± SD.

### 4.10. Statistical Analysis

The data (*n* = 3–6) in the present study was required to avoid uncertainty. All results are demonstrated as mean ± SD, as analyzed through SPSS.Statistics.v 17.0 software (SPSS Inc., Chicago, IL, USA). The significant differences among different treatments were analyzed via the one-way ANOVA method. A *p* value < 0.05 was considered to be a significant difference.

## 5. Conclusions

In the present research, we prepared various concentrations of AuNP solutions (1.25, 2.5, 5, and 10 ppm) in order to investigate the biocompatibility and biological performance of MSCs. According to the in vitro results, AuNP solutions at 1.25 and 2.5 ppm significantly induced MSC proliferation and inhibited macrophage activation as well. Moreover, MSC morphology, including filopodia and lamellipodia, could be clearly observed under the AuNP 1.25 and 2.5 ppm treatments. The biological functions, e.g., MMP-2/9 expression and cell migration ability, were also remarkably enhanced with AuNP 1.25 and 2.5 ppm treatments, while anti-apoptotic proteins and IL-10 anti-inflammation cytokines were also highly expressed. Furthermore, the in vivo rat model investigations elucidated that foreign body responses and inflammations were significantly decreased through AuNP 1.25 and 2.5 ppm treatments, while endothelialization ability was also facilitated. In conclusion, we considered the higher concentration (5 and 10 ppm) of AuNP may cause the heterogeneity morphology for biomaterial modification, which can influence the adhesion capacity of MSCs. On the contrary, the AuNP at 1.25 and 2.5 ppm are suggested as the optimal concentration for biomaterial modification to induce better biocompatibility and biological functions such as adhesion, proliferation, anti-inflammation and MMPs secretion within MSCs. This study exhibits that the physically synthesized AuNPs at appropriate concentrations of 1.25 and 2.5 ppm can be a promising nano-platform coupling with MSCs therapeutic strategies for tissue regeneration.

## Figures and Tables

**Figure 1 ijms-24-00005-f001:**
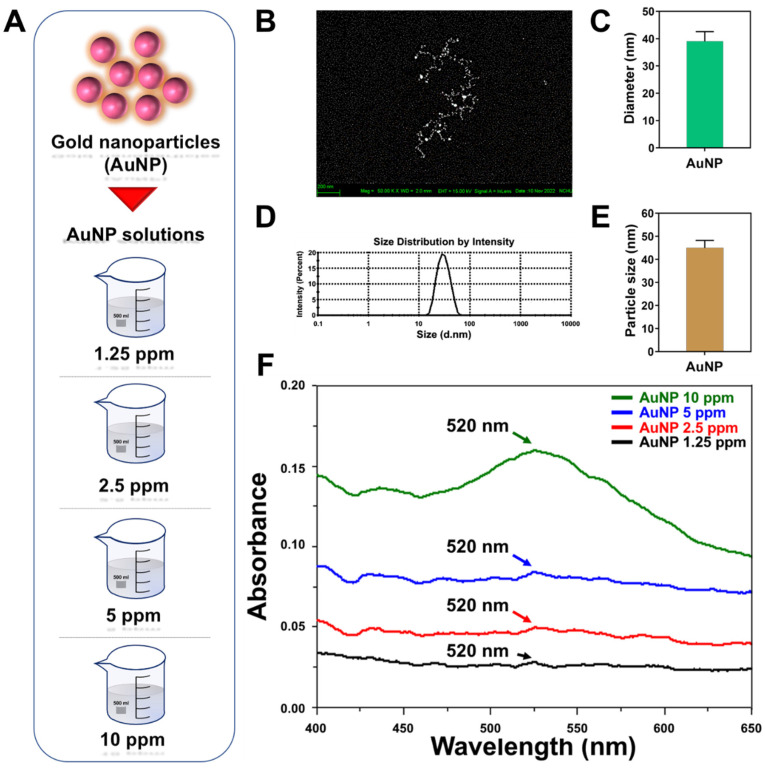
AuNP characterization. (**A**) The brief procedure to prepare various concentrations of AuNP solutions. The concentrations included 1.25, 2.5, 5 and 10 ppm. (**B**,**C**) The shape of the AuNPs was visualized by SEM, while the scale bar was set as 200 nm. The diameter of the AuNPs was quantified with Image J software, which was presented as 39 ± 3.5 nm. (**D**,**E**) The hydrodynamic size distribution intensity of the AuNPs was evaluated through DLS assay. The AuNP particle size was measured based on DLS assay, which was indicated as 45 ± 3.2 nm. (**F**) The as-prepared AuNP solutions with different amounts were subjected to UV-Vis assay. The spectra demonstrated the wavelength at 520 nm with the presence of AuNPs, while a higher concentration of AuNPs led to a stronger absorbance peak. The above results are exhibited as one of three independent experiments.

**Figure 2 ijms-24-00005-f002:**
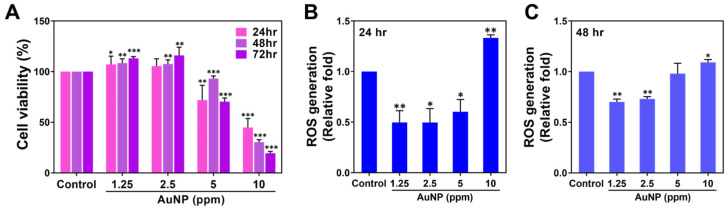
Cell viability and ROS generation of MSCs treated with various concentrations of AuNPs. (**A**) The cell viability of MSCs treated with AuNPs was investigated. The results indicate that the AuNP 1.25 and 2.5 ppm treatments enhanced better MSCs proliferation at 48 and 72 h, while the AuNP 5 and 10 ppm treatments had significantly lower cell viability at each time point. (**B**,**C**) The intracellular ROS generation in the MSCs was detected using the FACS method after 24 and 48 h of treatment. The results demonstrate that the AuNP 1.25 and 2.5 ppm groups induced lower ROS production at both time points. The results are displayed as mean ± SD (*n* = 3). * *p* < 0.05, ** *p* < 0.01, *** *p* < 0.001: compared to the Control group (TCPS).

**Figure 3 ijms-24-00005-f003:**
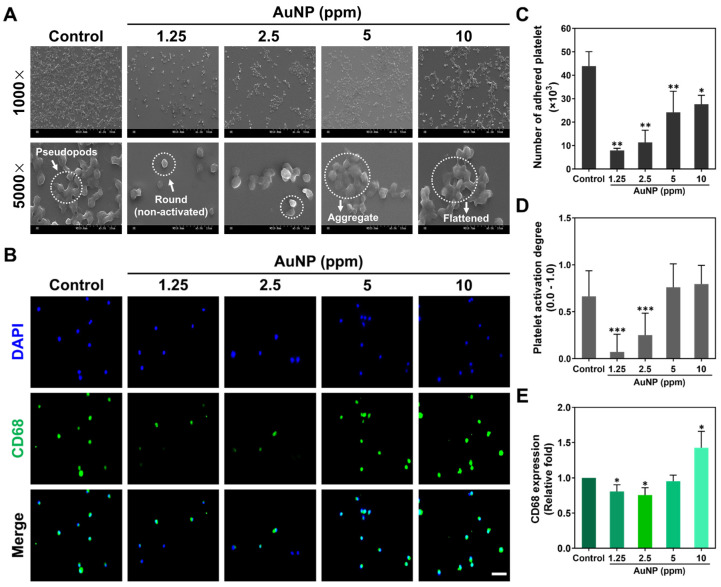
Activation of platelet cells and monocytes after AuNP treatments. (**A**) The morphology of platelets after 24 h of AuNP treatments was explored via SEM. The white arrow in the Control group is represented as pseudopods morphology. The round shape in the 1.25 ppm AuNP group indicates a non-activated form, while the white arrow in the 5 ppm AuNP treatment indicates an aggregated state. (**B**) A CD68 macrophage marker expressed in monocytes is demonstrated using the IF staining method after 96 h of treatment. Blue fluorescence is the cell nuclei located by a DAPI solution, while green fluorescence indicates the expression of CD68. Scale bar = 20 μm. (**C**,**D**) The adhered number and activation degree of platelet cells were quantified for each treatment. The results elucidate that the 1.25 and 2.5 ppm AuNP groups had the lower platelets adhered amount and activation degree. (**E**) The fluorescence intensity of CD68 expression was semi-quantified, which demonstrated the lower expression in both 1.25 and 2.5 ppm AuNP treatments. Scale bar = 50 μm. The above results are represented as mean ± SD (*n* = 3). * *p* < 0.05, ** *p* < 0.01, *** *p* < 0.001: compared to the Control group (TCPS).

**Figure 4 ijms-24-00005-f004:**
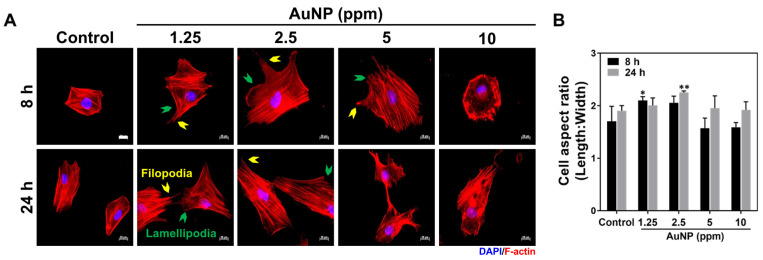
Cell morphology of MSCs influenced by AuNPs after 8 and 24 h of incubation. (**A**) The F-actin fiber of MSCs was stained by rhodamine phalloidin (red fluorescence) in each treatment. Yellow arrows indicate filopodia, while green arrows represent lamellipodia. DAPI was applied for cell nuclei detection. Scale bar = 20 μm. (**B**) The quantitative results of the cell aspect ratio for each treatment. The images are displayed as one of three independent experiments. The quantified results are exhibited as mean ± SD (*n* = 3). * *p* < 0.05, ** *p* < 0.01: compared to the Control group (TCPS).

**Figure 5 ijms-24-00005-f005:**
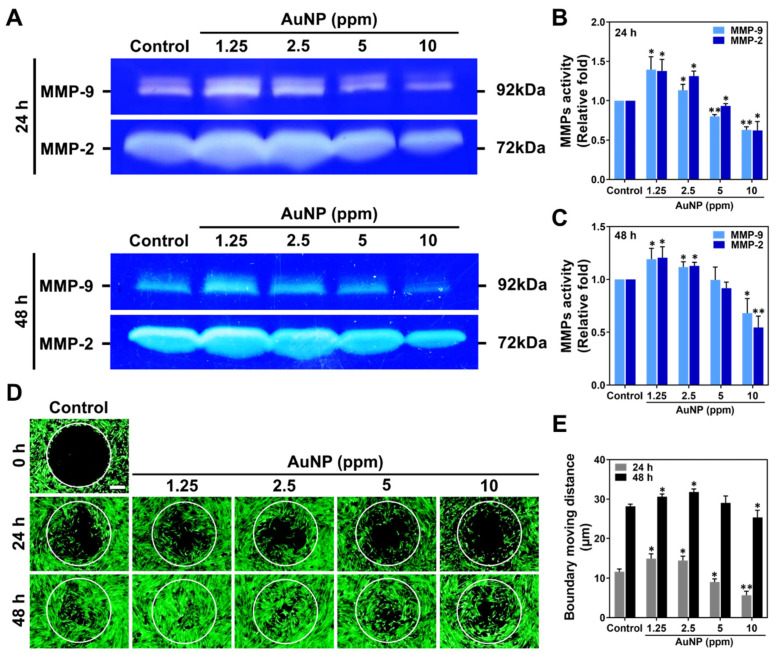
Metalloproteinase activities and cell migration ability of MSCs stimulated by AuNPs. (**A**) The zymography of MMP-2/9 enzymatic activities at 24 and 48 h are shown. (**B**,**C**) The expression level of MMP-2/9 at 24 and 48 h was semi-quantified. Both MMP-2 and MMP-9 protein expression were significantly increased in 1.25 and 2.5 ppm AuNP treatments. (**D**) The fluorescence images of the MSCs’ migration ability are displayed. Scale bar = 200 μm. (**E**) The boundary moving distance was measured, and the result explains that the 1.25 and 2.5 ppm AuNP groups remarkably strengthened the moving distance of MSCs. The images were exhibited from one of three independent experiments. The semi-quantitative data are shown as mean ± SD (*n* = 3). * *p* < 0.05, ** *p* < 0.01: compared to the Control group (TCPS).

**Figure 6 ijms-24-00005-f006:**
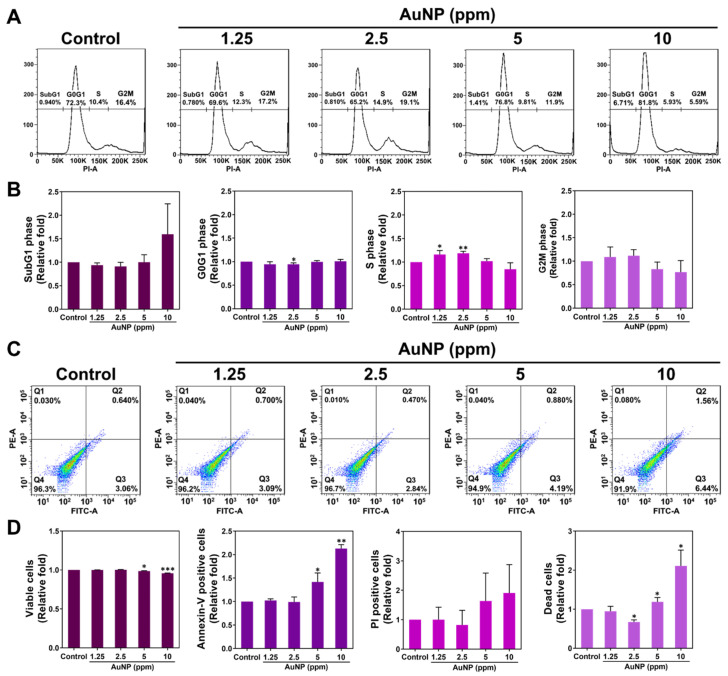
Cell cycle progress and Apoptosis of MSCs with AuNP treatments. (**A**) The cell cycle histograms of MSCs were acquired through the FACS method. (**B**) The relative fold of the Sub-G1 phase, G0G1 phase, S phase and G2M phase were quantified. The results indicate no significant difference at the Sub-G1 phase in each experimental group except for the higher ratio in the 10 ppm AuNP treatment. Furthermore, the ratio at the S phase was significantly higher in both 1.25 and 2.5 ppm AuNP groups, demonstrating the enhancement of cell proliferation. (**C**) Apoptotic MSCs were also detected via the FACS method. (**D**) The viable cells, apoptotic cells and dead cells were semi-quantified. The data of Annexin-V positive cells (apoptosis) evaluated there was no significant difference when treated with 1.25 and 2.5 ppm AuNPs. On the contrary, 5 and 10 ppm AuNPs induced more apoptotic and dead cells. The histograms were chosen from one of three independent experiments. Semi-quantifications are evaluated as mean ± SD (*n* = 3). * *p* < 0.05, ** *p* < 0.01, *** *p* < 0.001: compared to the Control group (TCPS).

**Figure 7 ijms-24-00005-f007:**
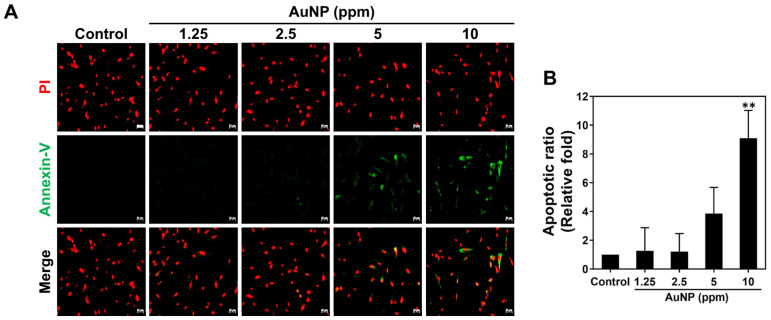
Apoptotic ratio of MSCs visualized via Annexin-V/PI double staining at 48 h. (**A**) The fluorescence images of apoptotic MSCs cultured with various concentrations of AuNPs are shown. (**B**) The apoptotic results in each treatment were semi-quantified based on the green fluorescence intensity, indicating no significant difference in both 1.25 and 2.5 ppm AuNP groups. Scale bars measured as 50 μm. The images were presented as one of three independent experiments. Data are analyzed as mean ± SD (*n* = 3). ** *p* < 0.01: compared to the Control group (TCPS).

**Figure 8 ijms-24-00005-f008:**
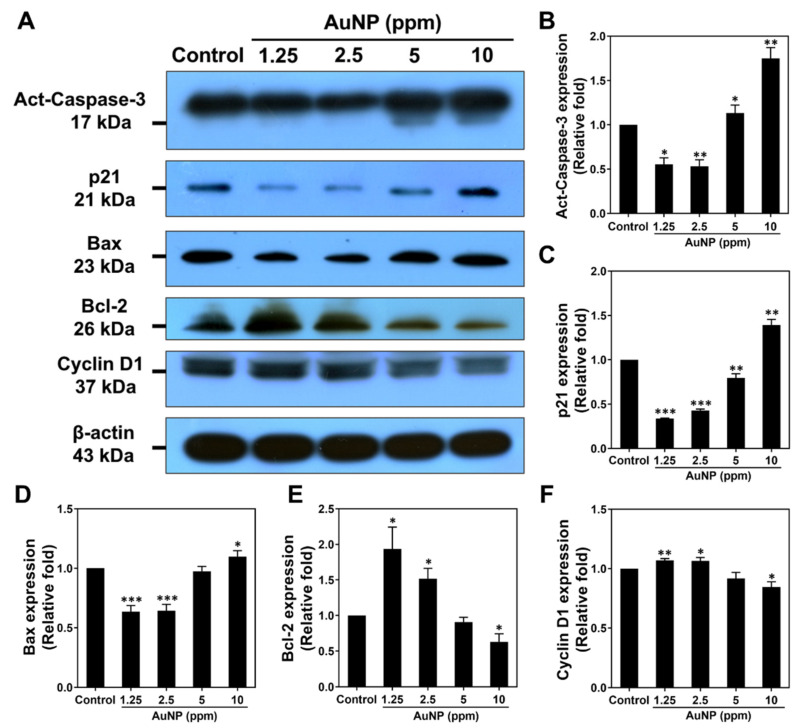
Measurement of apoptotic-related protein expression in MSCs at 48 h. (**A**) The immunoblots of each protein expressed in various AuNP treatments were exhibited. The protein expression was normalized to β-Actin. (**B**) Semi-quantification of active-caspase-3 apoptotic protein. (**C**) Semi-quantification of p21 protein. (**D**) Semi-quantification of Bax apoptotic protein. (**E**) Semi-quantification of Bcl-2 anti-apoptotic protein. (**F**) Semi-quantification of Cyclin-D1 protein. The images of immunoblots were demonstrated as one of three independent experiments. Protein expression levels were analyzed as mean ± SD (*n* = 3). * *p* < 0.05, ** *p* < 0.01, *** *p* < 0.001: compared to the Control group (TCPS).

**Figure 9 ijms-24-00005-f009:**
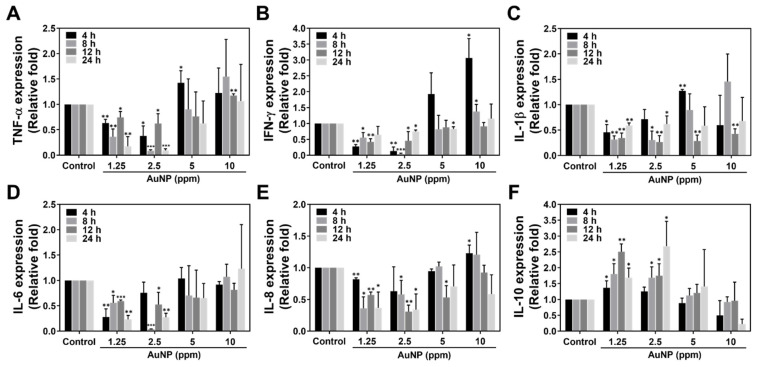
Expressed mRNA for inflammatory cytokines in MSCs. The mRNA expression of inflammatory-related cytokines: (**A**) TNF-α, (**B**) IFN-γ, (**C**) IL-1β, (**D**) IL-6, (**E**) IL-8 and (**F**) IL-10 were analyzed with real-time PCR assay at 4, 8, 12 and 24 h. In accordance with the semi-quantitative results, the expressed mRNA of TNF-α, IFN-γ, IL-1β, IL-6 and IL-8 inflammatory cytokines was significantly lower in the 1.25 and 2.5 ppm AuNP treatments. Moreover, the anti-inflammatory cytokine IL-10 was explored to be remarkably higher expressed in MSCs in both 1.25 and 2.5 ppm AuNP groups. The mRNA expression level was quantified as mean ± SD (*n* = 3). * *p* < 0.05, ** *p* < 0.01, *** *p* < 0.001: compared to the Control group (TCPS).

**Figure 10 ijms-24-00005-f010:**
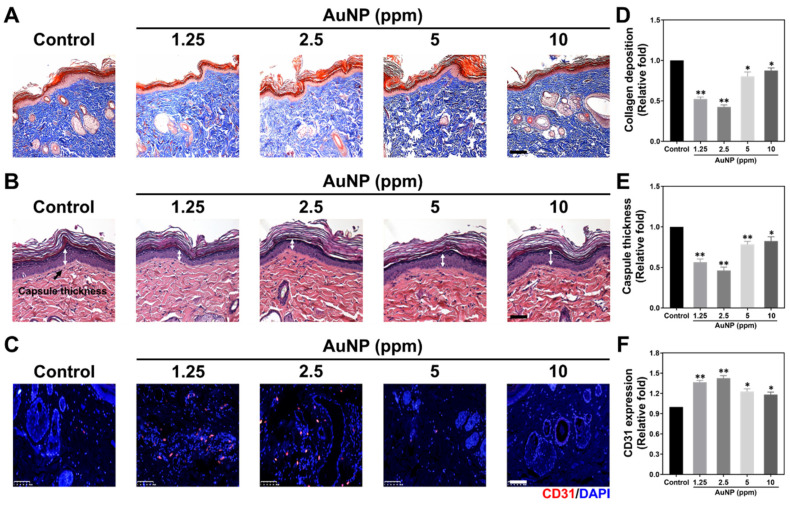
Biocompatibility investigations in the rat model with four weeks of AuNP treatments. (**A**) The collagen deposition (blue color) was measured via Masson’s trichrome staining assay. (**B**) The capsule formation (white double arrow) was explored through H&E staining assay. (**C**) The expression of the CD31 endothelial marker was observed after immunohistochemical (IHC) staining assay. The cell nuclei were located with DAPI solution (blue fluorescence), while the red fluorescence was presented as a CD31 marker expression. The scale bars of images were equal to 100 μm. (**D**) The semi-quantitative results of collagen deposition were influenced by AuNP treatments. (**E**) The capsule thickness of tissues in each treatment was semi-quantified. (**F**) The CD31 expression intensity was also quantified. The above evidence indicates that AuNPs at the concentrations of 1.25 and 2.5 ppm could induce the lowest foreign body response, and significantly enhanced the endothelialization for tissue regeneration. The data are demonstrated as mean ± SD (*n* = 3). * *p* < 0.05, ** *p* < 0.01: compared to the Control group (TCPS).

**Figure 11 ijms-24-00005-f011:**
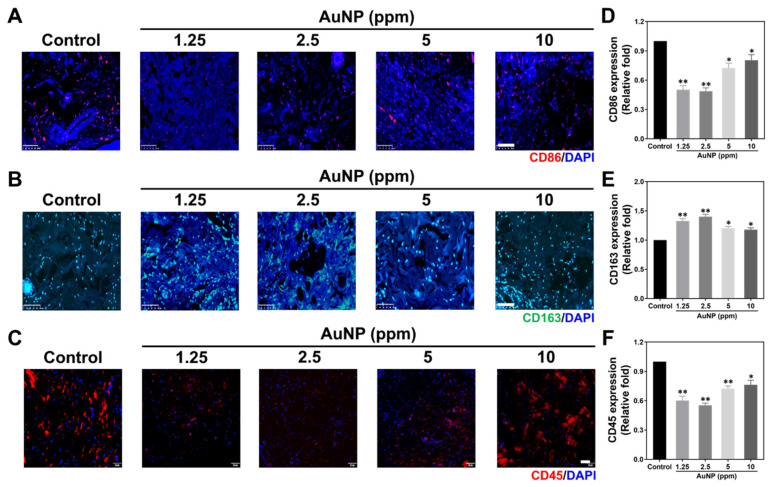
Evaluation of anti-inflammatory abilities in rat models with AuNP treatments for 4 weeks via IHC staining assay. The images of (**A**) CD86 marker expression (M1 macrophage polarization, red) and (**B**) CD163 marker expression (M2 polarization, green) are displayed. The scale bars were set at 100 μm. (**C**) The CD45 marker expression (red fluorescence) is presented as the leukocyte filtration influenced by AuNP treatments. The scale bar was measured as 50 μm. The (**D**) CD86 and (**E**) The CD163 macrophage marker’s expression intensity was semi-quantified. (**F**) The semi-quantified data of CD45 marker expression intensity. The results demonstrate that 1.25 and 2.5 ppm AuNP treatments have good biocompatibility in an animal model. The data are demonstrated as mean ± SD (*n* = 3). * *p* < 0.05, ** *p* < 0.01: compared to the Control group (TCPS).

**Figure 12 ijms-24-00005-f012:**
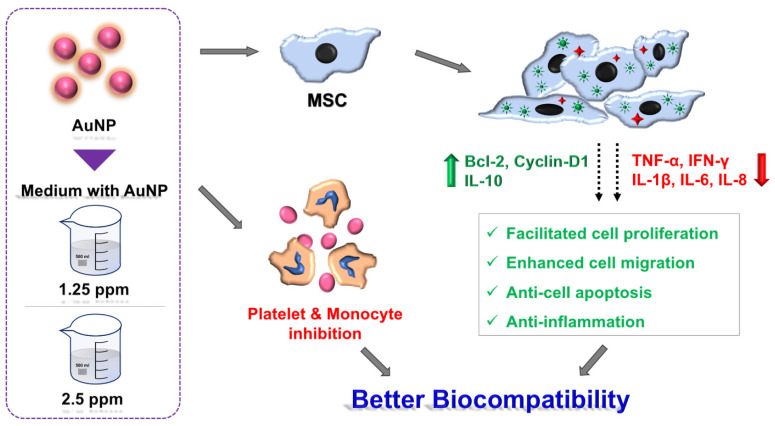
Schematic illustration. In line with our investigations, AuNP at the concentrations of 1.25 and 2.5 ppm facilitated better biocompatibility and biological performance, with the enhancement of cell proliferation, migration and anti-immune response capacities. Additionally, the 1.25 and 2.5 ppm AuNP treatments also prevented MSCs from apoptosis. In conclusion, 1.25 and 2.5 ppm AuNPs are suggested to be better concentrations for the biomedical approach in drug delivery or nanomaterial modification.

## Data Availability

Data are contained within the article.

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
