# Peer review of "Favorable Biological Performance Regarding the Interaction between Gold Nanoparticles and Mesenchymal Stem Cells"

_ijms, 2022, doi:10.3390/ijms24010005_

Round 1

Reviewer 1 Report

Manuscript title: Favorable Biological Performance Regarding the Interaction Between Gold Nanoparticles and Mesenchymal Stem Cells

Journal: International Journal of Molecular Sciences

The gold nanoparticles and mesenchymal stem cells are important subject which have a great potential for investigation. There are few points in the manuscript which should be clarified and broadened before acceptance.

  • The authors should state more clearly which are the novelties of their work and highlight the novel experimental results in comparison with literature data.
  • Introduction is appropriate and covers important studies published elsewhere. However, the need for this study is not explained well.

o   Authors said:” A Scanning electron microscope was applied to observe the AuNPs used in the present research (Figure 1B), while the right panel displayed the image at higher magnification.” Authors should better analyzed the micrographs.

o   Authors said:” According to the in vitro results, AuNP solutions at 1.25 and 2.5 ppm significantly induced MSC proliferation, and inhibited macrophage activation as well.” Authors should better explain this statement in comparison with results of other AuNP solutions.

o   Authors said:” …..MMP-2/9 expression and cell migration ability, were also remarkably enhanced with AuNP 1.25 and 2.5 ppm treatments….” Authors should give better discussion on this subject.

o   The cell cycle progress and Apoptosis of MSCs with AuNP treatments and measurement of apoptotic-related protein expression in MSCs at 48 hours are quite interesting and need proper attention for more explanation (Figs. 6 and 8).

o   Authors should better explain what is needed for practical applications of the samples.  

Author Response

Response letter

Ms ID: ijms-2071382 manuscript R1

Title:Favorable Biological Performance Regarding the Interaction Between Gold Nanoparticles and Mesenchymal Stem Cells

It is very much appreciate that the reviewer made these value comments. We have attended to the comments and made the revision accordingly.

Reviewer(s)' Comments to Author:

Reviewer 1:

Manuscript title: Favorable Biological Performance Regarding the Interaction Between Gold Nanoparticles and Mesenchymal Stem Cells

Journal: International Journal of Molecular Sciences 

The gold nanoparticles and mesenchymal stem cells are important subject which have a great potential for investigation. There are few points in the manuscript which should be clarified and broadened before acceptance.

  1. The authors should state more clearly which are the novelties of their work and highlight the novel experimental results in comparison with literature data.
    Answer:
    Thanks for the valuable suggestion from the reviewer. We have included the more detail description of novel perspective related to this study in the “Discussion” section.
    (1) “Various approaches for synthesis of AuNP including chemical and physical techniques have been investigated for decades [40]. The chemical synthesis of AuNP can be easily functionalized for specific applications [41]. However, the usage of reducing agent to acquire AuNP and the functionalization via organic solvents and reagents are high-priced and toxic for biomedical applications and environment [42]. Moreover, the production of AuNP via chemical process is low, which causes limitation for investigations [43]. In comparison with chemical manufacturing methods, the physical synthesis AuNP is acquired from physical vapor deposition technique without addition of surfactant, stabilizer and reducing agents, which are characterized to be controlled size at nano-scale, high surface area to volume ratio and superior biocompatibility with low toxicity for targeted drug delivery applications [40, 44]. Thus, the physical synthesis spherical AuNPs were applied for investigations of biological performances within MSCs.” (Page 14, Line 451-463)
    (2) “As mentioned with a review report, the conventional approaches to synthesis AuNP that combined with biomaterials may decrease the durability and cause toxicity or side effects that reduce therapeutic efficiency [60]. Thus, the physically synthesis AuNPs at an appropriate concentration in this study were the promising candidate for biomedical applications. While the particle surface could be functionalized by biopolymer such as collagen or polyethylene glycol, which were applied as the biocompatible nanodrug delivery system with superior stability [20, 61]. In accordance with this study, the physical synthesis AuNPs fabricated polymeric biomaterials have more potential for clinical applications with good biosafety owing to the minimal toxicity effects without using reducing agents. Moreover, the functionalized AuNP to carry various biomolecules can be manufactured for investigations of facilitating specific differentiation pathways in MSCs with better tissue repair.” (Page 15, Line 518-529)

  1. Introduction is appropriate and covers important studies published elsewhere. However, the need for this study is not explained well.

Answer:
Thanks for the valuable comment. We have included more detail decription in the “Introduction” section. “As stated above, in view of the advantages of physical manufactured AuNPs, the biocompatible AuNPs have been used for modification of biomaterials to achieve better biological efficiency [9, 12, 25]. In the present study, we applied MSCs to use as treatment with various concentrations of AuNP in order to evaluate their biological performance, with expectations of determining optimal AuNP concentration for enhancing drug delivery and tissue engineering efficiency in clinical approaches.” (Page 3, Line 129-134)

  1. Authors said:” A Scanning electron microscope was applied to observe the AuNPs used in the present research (Figure 1B), while the right panel displayed the image at higher magnification.” Authors should better analyzed the micrographs.

Answer:
Thanks for the kind of suggestion. We have included the new SEM image for AuNP observation in the Figure 1B, with the diameter was quantified in the Figure 1C by Image J software.
(1) “A Scanning electron microscope was applied to observe the AuNPs used in the present research (Figure 1B), while the AuNP diameter quantified via Image J software indicates as 39 ± 3.5 nm (Figure 1C). Figure 1D demonstrates the histogram of particle size distribution measured by DLS assay, and the accurate diameter of the AuNPs was measured as 45 ± 3.2 nm (Figure 1E).” (Page 3, Line 140-144)
(2) “(B, C) The shape of AuNP was visualized by SEM, while the scale bar was set as 200 nm. The diameter of AuNP was quantified with Image J software, which was presented as 39 ± 3.5 nm. (D, E) The hydrodynamic size distribution intensity of AuNP was evaluated through DLS assay.” (Page 4, Line 151-154)

  1. Authors said:” According to the in vitro results, AuNP solutions at 1.25 and 2.5 ppm significantly induced MSC proliferation, and inhibited macrophage activation as well.” Authors should better explain this statement in comparison with results of other AuNP solutions.
    Answer:
    Thanks for the kindly suggestion from the reviewer. We have included the more detail explanation in the conclusion.
    “…In conclusion, we considered the higher concentration (5 and 10 ppm) of AuNP may cause the heterogeneity morphology for biomaterial modification, which can influence the adhesion capacity of MSCs. On the contrary, the AuNP at 1.25 and 2.5 ppm are considered as the optimal concentration for biomaterial modification to induce better biocompatibility and biological functions such as adhesion, proliferation, anti-inflammation and MMPs secretion within MSCs...” (Page 20, Line 768-773)

  1. Authors said:” …..MMP-2/9 expression and cell migration ability, were also remarkably enhanced with AuNP 1.25 and 2.5 ppm treatments….” Authors should give better discussion on this subject.
    Answer:
    Thanks for the valuble suggestion. We have added the more detail description in the “Discussion” section.

(1) “…As literature reported, SDF-1/CXCR4 axis initiate cell signaling pathways in cells associated with secreting matrix metalloproteinases (MMPs) and vascular endothelial growth factor (VEGF) [52, 53]. Afterwards, the MMPs would be expressed in cells to degrade ECM that associated with cell migration and vascular remodeling [54], while the MMP-2 and MMP-9 proteins could activate α5β3 integrin to enhance migratory ability and angiogenesis for tissue repair [55]…” (Page 15, Line 492-497)
(2) “Thus, 1.25 to 2.5 ppm of AuNP may be the appropriate concentration for future modified nanomaterials to have homogenous morphology for inducing better adhesion and MMPs secretion of mesenchymal stem cells.” (Page 15, Line 505-507)

  1. The cell cycle progress and Apoptosis of MSCs with AuNP treatments and measurement of apoptotic-related protein expression in MSCs at 48 hours are quite interesting and need proper attention for more explanation (Figs. 6 and 8).
    Answer:
    Thanks for the valuable comment from the reviewer. We have included more detail explantion focused on cell cycle progress and apoptosis induction in the “Results 2.4.” section.
    (1) “…The cell progress at G0G1 quantification showed a slightly decreased with 1.25 and 2.5 ppm AuNP addition. Interestingly, we discovered that the cell population at S phase was greater in 1.25 and 2.5 ppm AuNP addition with significant difference, indicating the MSCs passed G1 phase and entered S phase that associated with mitosis and proliferation efficiency. However, the AuNP 5 treatment did not facilitate more MSCs to enter S phase, moreover, the decreased cell population was explored in AuNP 10 ppm. For G2M phase, the quantified data demonstrated the higher level in 1.25 and 2.5 ppm AuNP groups. On the contrary, the lower MSCs population was found in 5 and 10 ppm treatments. The evidence elucidated that at the concentration of 1.25 and 2.5 ppm, AuNP could facilitate MSCs to undergo S phase and enter G2M cell cycle progress….” (Page 8, Line 265-275)
    (2) “…We analyzed the quantitative results with the exploration of higher Annexin-V positive cells population (presented as apoptotic cell) induced by AuNP at the concentration of 5 and 10 ppm, while the similar trend also shared in the detection of PI positive cells (presented as necrotic cells). The results of viable MSCs demonstrated the decreased level with significant difference in AuNP 5 and 10 ppm. On the contrary, the treatment of AuNP 1.25 and 2.5 ppm did not induce higher Annexin-V positive and PI positive MSCs (Figure 6D). Discussed with the viable and dead MSCs stimulated by each treatment, the results elucidated that the AuNP 2.5 and 5 ppm treatments lead to less viable cells and more dead cells, while the 1.25 and 2.5 ppm AuNP induced the opposite conditions (Figure 6D)….” (Page 8, Line 278-287)
    (3) “…In accordance with the data, the apoptotic relative proteins, including Active-caspase-3, p21and Bax, were determined to have significantly lower expression in both AuNP 1.25 and 2.5 ppm treatments. We explored that the treatment of AuNP 5 and 10 ppm induced greater apoptotic proteins expression, particularly for AuNP 10 ppm with significant difference. On the contrary, the apoptosis inhibited protein Bcl-2 and cell cycle progress regulator protein Cyclin D1 were investigated to be remarkably expressed in MSCs in both AuNP 1.25 and 2.5 ppm treatments. Additionally, in the treatment of AuNP 5 ppm, both Bcl-2 and Cyclin D1 expression were slightly decreased, while AuNP 10 ppm group showed even lower expression with significant difference….” (Page 8, Line 298-306)

  1. Authors should better explain what is needed for practical applications of the samples.  
    Answer:
    Thanks for the kindly comment. We have included the future applications of AuNPs in the “Discussion” section and Conclusion.
    (1) “..As mentioned with a review report, the conventional approaches to synthesis AuNP that combined with biomaterials may decrease the durability and cause toxicity or side effects that reduce therapeutic efficiency [60]. Thus, the physically synthesis AuNPs at an appropriate concentration in this study were the promising candidate for biomedical applications. While the particle surface could be functionalized by biopolymer such as collagen or polyethylene glycol, which were applied as the biocompatible nanodrug delivery system with superior stability [20, 61]. In accordance with this study, the physical synthesis AuNPs fabricated polymeric biomaterials have more potential for clinical applications with good biosafety owing to the minimal toxicity effects without using reducing agents. Moreover, the functionalized AuNP to carry various biomolecules can be manufactured for investigations of facilitating specific differentiation pathways in MSCs with better tissue repair.” (Page 15, Line 518-529)
    (2) “This study exhibits that the physically synthesis AuNPs at the appropriate concentrations of 1.25 and 2.5 ppm are a promising nano-platform coupling with MSCs therapeutic strategies for tissue regeneration.” (Page 20, Line 774-776)

Reviewer 2 Report

In this article, Lin and co-workers delivered a work on gold nanoparticles for strengthening the biological function of MSCs. The synthesized AuNP s induced a lower expression of inflammatory cytokines and exhibited good biocompatibility performance towards regenerative research. This work is sounding. It is suggested for publication after a revision of some concerns.

1.     In Figure 1E, it seemed that there is no prominent absorption of AuNPs at 1.25 ppm. I suggest the authors provide a wider range of absorption wavelength.

2.     How about the long-term stability of AuNPs in biological medium?

3.     In addition to cytokines, whether AuNPs affect the body weight of mice? Also, H&E staining of normal organs should be provided to show its low systemic toxicity.

4.     The annotation for Figure 1B and Figure 1C is inappropriate. Please double check it.

5. It is suggested to conduct in vitro ROS staining (DCFH-DA) on MSCs to demonstrate the ROS generation capability of AuNPs.

Author Response

Response letter

Ms ID: ijms-2071382 manuscript R1

Title:Favorable Biological Performance Regarding the Interaction Between Gold Nanoparticles and Mesenchymal Stem Cells

It is very much appreciate that the reviewer made these value comments. We have attended to the comments and made the revision accordingly.

Reviewer(s)' Comments to Author:

Reviewer 2:

In this article, Lin and co-workers delivered a work on gold nanoparticles for strengthening the biological function of MSCs. The synthesized AuNP s induced a lower expression of inflammatory cytokines and exhibited good biocompatibility performance towards regenerative research. This work is sounding. It is suggested for publication after a revision of some concerns.

  1. In Figure 1E, it seemed that there is no prominent absorption of AuNPs at 1.25 ppm. I suggest the authors provide a wider range of absorption wavelength.

Answer:
Thanks for the kindly comment. Due to the concentration of AuNP (1.25 ppm) is very lower, the typical peak at 520 nm was not significant in the UV-Vis spectra. However, along with the higher concentration of 2.5 ppm, 5 ppm and 10 ppm, the absorption peaks at 520 nm become stronger and in a dose dependent manner. The wider range of absorption wavelength for each concentration of AuNP is demonstrated following by reviewer’s suggestion.

  1. How about the long-term stability of AuNPs in biological medium?

Answer:
We have performed and immersed the materials in the medium for 0 day (without immersion) or 7 days, and then cultured the cells on the materials for 48 hr and compared the growth effect of the MSCs on different materials by MTT assay for those without immersion and those immersed for 7 days. The growth efficiency of MSCs after 7 days of immersion compared with the results of 0 day of immersion showed a slight decrease in cell proliferation, but there was no significant difference. Therefore, we believed that the material properties did not change significantly during the 7 days of incubation.

Figure. MSC proliferation examined for materials immersed in the medium for 0 day and 7 days. MSCs were incubated on different materials and then subjected to MTT assay.  

  1. In addition to cytokines, whether AuNPs affect the body weight of mice? Also, H&E staining of normal organs should be provided to show its low systemic toxicity.

Answer:
Thanks for the valuable comment from the reviewer.
(1) The AuNPs were injected into our mice model, while the result showed no significant change of body weight between control group and AuNP treatment groups.

(2) The H&E staining for each organ in mice model was conducted after retro-orbital sinus injection of AuNP. The results of tissue integrity shows no significant destruction in each organ. We hope the reviewer can realize that this data will be publish in our another study.

Figure. Investigation of tissue integrity after 24 hours of Au treatment. The tissue integrity within brain, heart, liver, spleen, lung and kidney were evaluated by H&E staining assay. The tissue morphology of brain, heart, liver was observed with the scale bar setting as 5 mm and 100 μm. The tissue morphology of spleen, lung and kidney was revealed. Scale bars were measured as 2.5 mm and 100 μm.

  1. The annotation for Figure 1B and Figure 1C is inappropriate. Please double check it.

Answer:
Thanks for the valuable comment from the reviewer. We have corrected the figure caption for new Figure 1B-E with new SEM image for AuNP and diameter quantification by Image J software. The new description was included in the “Results 2.1.” section.
“A Scanning electron microscope was applied to observe the AuNPs used in the present research (Figure 1B), while the AuNP diameter quantified via Image J software indicates as 39 ± 3.5 nm (Figure 1C). Figure 1D demonstrates the histogram of particle size distribution measured by DLS assay, and the accurate diameter of the AuNPs was measured as 45 ± 3.2 nm (Figure 1E).” (Page 3, Line 140-144)
“(B, C) The shape of AuNP was visualized by SEM, while the scale bar was set as 200 nm. The diameter of AuNP was quantified with Image J software, which was presented as 39 ± 3.5 nm. (D, E) The hydrodynamic size distribution intensity of AuNP was evaluated through DLS assay.” (Page 4, Line 151-154)

  1. It is suggested to conduct in vitro ROS staining (DCFH-DA) on MSCs to demonstrate the ROS generation capability of AuNPs.

Answer:
Thanks for the valuable comment from the reviewer. We have conducted the ROS staining in MSCs by using DCFH-DA (2′,7′-dichlorofluorescin diacetate) fluorescent probe, and the fluorescein-positive cells were detected by FACS Calibur flow cytometer to quantify the ROS generation in each treatment. The results are displayed in Figure 2B-C, while the experimental procedure was described in the “section 4.4.1.”.
“Intercellular ROS production in MSCs with different treatments was detected by a DCFH-DA (2′,7′-dichlorofluorescin diacetate) fluorescent probe (Sigma-Aldrich, USA) [24]. The MSCs (2 × 105 cells/well) were seeded into a 6-well culture plate (37°C, 5% CO2) to stand for cell attachment after 24 hours. Next, the as-prepared AuNP solutions were loaded into the culture plate for 24 and 48 hours of incubation. After the incubation, a 0.05% Trypsin-EDTA solution was added to harvest the MSCs which were then washed with PBS. A DCFH-DA (10 nM) solution was used to detect ROS in a dark room for a period of 30 minutes. The fluorescein-positive cells were then examined by a flow cytometer and quantified using Flow Jo Version 7.6 software (Becton Dickinson, Canton, MA, USA).” (Page 16-17, Line 583-592)

Round 2

Reviewer 1 Report

The revised manuscript can be accepted in IJMS. 

Reviewer 2 Report

I agree to publish.